# Decomposition-Invariant Conditional Gradient for General Polytopes with Line Search

**Mohammad Ali Bashiri**          **Xinhua Zhang**

Department of Computer Science, University of Illinois at Chicago

Chicago, Illinois 60607

{mbashi4,zhangx}@uic.edu

## Abstract

Frank-Wolfe (FW) algorithms with linear convergence rates have recently achieved great efficiency in many applications. Garber and Meshi (2016) designed a new decomposition-invariant pairwise FW variant with favorable dependency on the domain geometry. Unfortunately it applies only to a restricted class of polytopes and cannot achieve theoretical and practical efficiency at the same time. In this paper, we show that by employing an away-step update, similar rates can be generalized to *arbitrary* polytopes with strong empirical performance. A new "condition number" of the domain is introduced which allows leveraging the sparsity of the solution. We applied the method to a reformulation of SVM, and the linear convergence rate depends, for the first time, on the number of support vectors.

## 1 Introduction

The Frank-Wolfe algorithm [FW, 1] has recently gained revived popularity in constrained convex optimization, in part because linear optimization on many feasible domains of interest admits efficient computational solutions [2]. It has been well known that FW achieves $O(1/\epsilon)$ rate for smooth convex optimization on a compact domain [1, 3, 4]. Recently a number of works have focused on linearly converging FW variants under various assumptions.

In the context of convex feasibility problem, [5] showed linear rates for FW where the condition number depends on the distance of the optimum to the relative boundary [6]. Similar dependency was derived in the local linear rate on polytopes using the away-step [6, 7]. With a different analysis approach, [8–10] derived linear rates when the Robinson's condition is satisfied at the optimal solution [11], but it was not made clear how the rate depends on the dimension and other problem parameters.

To avoid the dependency on the location of the optimum, [12] proposed a variant of FW whose rate depends on some geometric parameters of the feasible domain (a polytope). In a similar flavor, [13, 14] analyzed four versions of FW including away-steps [6], and their affine-invariant rates depend on the pyramidal width (Pw) of the polytope, which is hard to compute and can still be ill-conditioned. Moreover, [15] recently gave a duality-based analysis for non-strongly convex functions. Some lower bounds on the dependency of problem parameters for linear rates of FW are given in [12, 16].

To get around the lower bound, one may tailor FW to specific objectives and domains (*e.g.* spectrahedron in [17]). [18] specialized the pairwise FW (PFW) to simplex-like polytopes (SLPs) whose vertices are binary, and is defined by equality constraints and $x_i \geq 0$. The advantages include: a) the convergence rate depends linearly on the cardinality of the optimal solution and the domain diameter square ($D^2$), which can be much better than the pyramidal width; b) it is decomposition-invariant, meaning that it does not maintain a pool of atoms accumulated and the away-step is performed on the face that the current iterate lies on. This results in considerable savings in computation and storage.

| | PFW-1 [18] (SLP) | PFW-2 [18] general | LJ [13] general | AFW-1 (SLP) | AFW-2 general |
|---|---|---|---|---|---|
| Unit cube $[0,1]^n$ | $ns$ | $\times$ | $n^2$ | $ns$ | $n^2 s$ |
| $\mathcal{P}_k = \{\mathbf{x} \in [0,1]^n : \mathbf{1}^\top \mathbf{x} = k\}$ | $ks$ | $\times$ | $n\ (k=1)$ | $ks$ | $k^2 s$ |
| $\mathcal{Q}_k = \{\mathbf{x} \in [0,1]^n : \mathbf{1}^\top \mathbf{x} \le k\}$ | $\times$ | $\times$ | $k \cdot \mathrm{Pw}^{-2}$ | $\times$ | $k^2 \min(sk, n)$ |
| arbitrary polytope in $\mathbb{R}^n$ | $\times$ | $\times$ | $D^2 \cdot \mathrm{Pw}^{-2}$ | $\times$ | $D^2 n H_s$ |

Table 1: Comparison of related methods. These numbers need to be multiplied with $\kappa \log \frac{1}{\epsilon}$ to get the convergence rates, where $\kappa$ is the condition number of the objective, $D$ is the diameter of the domain, $s$ is the cardinality of the optimum, and Pw is the pyradimal width.. Our method is AFW. $\times$ means inapplicable or no rate known. PFW-1 [18] and AFW-1 apply only to SLP, hence not covering $\mathcal{Q}_k$ ($k \ge 2$). [13] showed the pyramidal width for $\mathcal{P}_k$ only with $k = 1$.

However, [18] suffers from multiple inherent restrictions. First it applies only to SLPs, which although encompass useful sets such as $k$-simplex $\mathcal{P}_k$, do not cover its convex hull with the origin ($\mathcal{Q}_k$):

$$\mathcal{P}_k = \{\mathbf{x} \in [0,1]^n : \mathbf{1}^\top \mathbf{x} = k\}, \quad \mathcal{Q}_k = \{\mathbf{x} \in [0,1]^n : \mathbf{1}^\top \mathbf{x} \le k\}, \quad \text{where } k \in \{1, \ldots, n\}.$$

Here $\mathbf{1} = (1, \ldots, 1)^\top$. Extending its analysis to general polytopes is not promising because it relies fundamentally on the integrality of the vertices. Second, its rate is derived from a delicately designed sequence of step size (PFW-1), which exhibits no empirical competency. In fact, the experiments in [18] resorted to line search (PFW-2). However no rate was proved for it. As shown in [13], dimension friendly bounds are intrinsically hard for PFW, and they settled for the factorial of the vertex number.

The goal of this paper is to address these two issues while at the same time retaining the computational efficiency of decomposition invariance. Our contributions are four folds. First we generalize the dimension friendly linear rates to arbitrary polytopes, and this is achieved by replacing the pairwise PFW in [18] with the away-step FW (AFW, §2), and setting the step sizes by line search instead of a pre-defined schedule. This allows us to avoid "swapping atoms" in PFW, and the resulting method (AFW-2) delivers not only strong empirical performance (§5) but also strong theoretical guarantees (§3.5), improving upon PFW-1 and PFW-2 which are strong in *either* theory *or* practice, but not both.

Second, a new condition number $H_s$ is introduced in §3.1 to characterize the dimension dependency of AFW-2. Compared with pyramidal width, it not only provides a more explicit form for computation, but also leverages the cardinality ($s$) of the optimal solution. This may lead to much smaller constants considering the likely sparsity of the solution. Since pyramidal width is hard to compute [13], we leave the thorough comparison for future work, but they are comparable on simple polytopes. The decomposition invariance of AFW-2 also makes each step much more efficient than [13].

Third, when the domain is indeed an SLP, we provide a step size schedule (AFW-1, §3.4) yielding the same rate as PFW-1. This is in fact nontrivial because the price for replacing PFW by AFW is the much increased hardness in maintaining the integrality of iterates. The current iterate is scaled in AFW, while PFW simply *adds* (scaled) new atoms (which on the other hand complicates the analysis for line search [13]). Our solution relies on first running a constant number of FW-steps.

Finally we applied AFW to a relaxed-convex hull reformulation of binary kernel SVM *with bias* (§4), obtaining $O(n\kappa(\#\mathrm{SV})^3 \log \frac{1}{\epsilon})$ *computational* complexity for AFW-1 and $O(n\kappa(\#\mathrm{SV})^4 \log \frac{1}{\epsilon})$ for AFW-2. Here $\kappa$ is the condition number of the objective, $n$ is the number of training examples, and #SV is the number of support vectors in the optimal solution. This is much better than the best known result of $O(n^3 \kappa \log \frac{1}{\epsilon})$ based on sequential minimal optimization [SMO, 19, 20], because #SV is typically much smaller than $n$. To the best of our knowledge, this is the first linear convergence rate for hinge-loss SVMs with bias where the rate leverages dual sparsity.

A brief comparison of our method (AFW) with [18] and [13] is given in Table 1. AFW-1 matches the superior rates of PFW-1 on SLPs, and AFW-2 is more general and its rate is slightly worse than AFW-1 on SLPs. PFW-2 has no rates available, and pyramidal width is hard to compute in general.

## 2    Preliminaries and Algorithms

Our goal is to solve $\min_{\mathbf{x} \in \mathcal{P}} f(\mathbf{x})$, where $\mathcal{P}$ is a polytope and $f$ is both strongly convex and smooth. A function $f : \mathcal{P} \to \mathbb{R}$ is $\alpha$-strongly convex if $f(\mathbf{y}) \ge f(\mathbf{x}) + \langle \mathbf{y} - \mathbf{x}, \nabla f(\mathbf{x}) \rangle + \frac{\alpha}{2} \|\mathbf{y} - \mathbf{x}\|^2$, $\forall \mathbf{x}, \mathbf{y} \in$

**Algorithm 1:** Decomposition-invariant Away-step Frank-Wolfe (**AFW**)

---

1  Initialize $\mathbf{x}_1$ by an arbitrary vertex of $\mathcal{P}$. Set $q_0 = 1$.

2  **for** $t = 1, 2, \ldots$ **do**

3       Choose the FW-direction via $\mathbf{v}_t^+ \leftarrow \arg\min_{\mathbf{v} \in \mathcal{P}} \langle \mathbf{v}, \nabla f(\mathbf{x}_t) \rangle$, and set $\mathbf{d}_t^{\text{FW}} \leftarrow \mathbf{v}_t^+ - \mathbf{x}_t$.

4       Choose the away-direction $\mathbf{v}_t^-$ by calling the away-oracle in (3), and set $\mathbf{d}_t^{\text{A}} \leftarrow \mathbf{x}_t - \mathbf{v}_t^-$.

5       **if** $\langle \mathbf{d}_t^{\text{FW}}, -\nabla f(\mathbf{x}_t) \rangle \geq \langle \mathbf{d}_t^{\text{A}}, -\nabla f(\mathbf{x}_t) \rangle$ **then** $\mathbf{d}_t \leftarrow \mathbf{d}_t^{\text{FW}}$, **else** $\mathbf{d}_t \leftarrow \mathbf{d}_t^{\text{A}}$. ▷ Choose a direction

6       Choose the **step size** $\eta_t$ by using one of the following two options:

7       **Option 1: Pre-defined step size:**    ▷ This is for SLP only. Need input arguments $n_0, \gamma_t$.

8       **if** $t \leq n_0$ **then**

9          Set $q_t = t$, $\eta_t = \frac{1}{t}$, and revert $\mathbf{d}_t = \mathbf{d}_t^{\text{FW}}$.    ▷ Perform FW-step for the first $n_0$ steps

10      **else**

11          Find the smallest integer $s \geq 0$ such that $q_t$ defined as follows satisfies $q_t \geq \lceil 1/\gamma_t \rceil$:

$$q_t \leftarrow \begin{cases} 2^s q_{t-1} + 1 & \text{if line 5 adopts the FW-step} \\ 2^s q_{t-1} - 1 & \text{if line 5 adopts the away-step} \end{cases}, \quad \text{and} \quad \eta_t \leftarrow q_t^{-1}. \quad (2)$$

13      **Option 2: Line search:** $\eta_t \leftarrow \arg\min_{\eta \geq 0} f(\mathbf{x}_t + \eta \mathbf{d}_t)$, $s.t.$ $\mathbf{x}_t + \eta \mathbf{d}_t \in \mathcal{P}$. ▷ General purpose

14      $\mathbf{x}_{t+1} \leftarrow \mathbf{x}_t + \eta_t \mathbf{d}_t$. Return $\mathbf{x}_t$ if $\langle -\nabla f(\mathbf{x}_t), \mathbf{d}_t^{\text{FW}} \rangle \leq \epsilon$.

---

**Algorithm 2:** Decomposition-invariant Pairwise Frank-Wolfe (**PFW**) (exactly the same as [18])

---

1  ... as in Algorithm 1, except replacing a) line 5 by $\mathbf{d}_t = \mathbf{d}_t^{\text{PFW}} := \mathbf{v}_t^+ - \mathbf{v}_t^-$, and b) line 8-11 by
    **Option 1: Pre-defined step size:** Find the smallest integer $s \geq 0$ such that $2^s q_{t-1} \geq 1/\gamma_t$.
               Set $q_t \leftarrow 2^s q_{t-1}$ and $\eta_t \leftarrow q_t^{-1}$. ▷ This option is for SLP only.

---

$\mathcal{P}$. In this paper, all norms are Euclidean, and we write vectors in bold lowercase letters. $f$ is $\beta$-smooth if $f(\mathbf{y}) \leq f(\mathbf{x}) + \langle \mathbf{y} - \mathbf{x}, \nabla f(\mathbf{x}) \rangle + \frac{\beta}{2} \|\mathbf{y} - \mathbf{x}\|^2, \forall \mathbf{x}, \mathbf{y} \in \mathcal{P}$. Denote the condition number as $\kappa = \beta/\alpha$, and the diameter of the domain $\mathcal{P}$ as $D$. We require $D < \infty$, *i.e.* the domain is bounded.

Let $[m] := \{1, \ldots, m\}$. In general, a polytope $\mathcal{P}$ can be defined as

$$\mathcal{P} = \{\mathbf{x} \in \mathbb{R}^n : \langle \mathbf{a}_k, \mathbf{x} \rangle \leq b_k, \ \forall k \in [m], \ C\mathbf{x} = \mathbf{d}\}. \quad (1)$$

Here $\{\mathbf{a}_k\}$ is a set of "directions" and is finite ($m < \infty$) and $b_k$ cannot be reduced without changing $\mathcal{P}$. Although the equality constraints can be equivalently written as two linear inequalities, we separate them out to improve the bounds below. Denoting $A = (\mathbf{a}_1, \ldots, \mathbf{a}_m)^\top$ and $\mathbf{b} = (b_1, \ldots, b_m)^\top$, we can simplify the representation into $\mathcal{P} = \{\mathbf{x} \in \mathbb{R}^n : A\mathbf{x} \leq \mathbf{b}, \ C\mathbf{x} = \mathbf{d}\}$.

In the sequel, we will find highly efficient solvers for a special class of polytope that was also studied by [18]. We call a potytope as a simplex-like polytope (SLP), if all vertices are binary (*i.e.* the set of extreme points $\text{ext}(\mathcal{P})$ are contained in $\{0, 1\}^n$, *and* the only inequality constraints are $\mathbf{x} \in [0, 1]^n$.[1]

Our decomposition-invariant Frank-Wolfe (FW) method with away-step is shown in Algorithm 1. There are two different schemes of choosing the step size: one with fixed step size (AFW-1) and one with line search (AFW-2). Compared with [13], AFW-2 enjoys decomposition invariance. Like [13], we also present a pairwise version in Algorithm 2 (PFW), which is exactly the method given in [18].

The efficiency of line search in step 13 of Algorithm 1 depends on the polytope. Although in general one needs a problem-specific procedure to compute the maximal step size, we will show in experiments some examples where such procedures with high computational efficiency are available.

The idea of AFW is to compute a) the FW-direction in the conventional FW sense (call it FW-oracle), and b) the away-direction (call it away-oracle). Then pick the one that gives the steeper descent and take a step along it. Our away-oracle adopts the decomposition-invariant approach in [18], which differs from [13] by saving the cost of maintaining a pool of atoms. To this end, our search space in the away-oracle is restricted to the vertices that satisfy all the inequality constraints by equality if the

current $\mathbf{x}_t$ does so:

$$\mathbf{v}_t^- := \arg\max_{\mathbf{v}} \langle \mathbf{v}, \nabla f(\mathbf{x}_t) \rangle, \ s.t. \ A\mathbf{v} \le \mathbf{b}, \ C\mathbf{v} = \mathbf{d}, \text{ and } \langle \mathbf{a}_i, \mathbf{x}_t \rangle = b_i \ \Rightarrow \ \langle \mathbf{a}_i, \mathbf{v} \rangle = b_i \ \forall i. \ (3)$$

Besides saving the space of atoms, this also dispenses with computing the inner product between the gradient and *all* existing atoms. Same as [18], it presumes efficient solutions to the away-oracle, which may preclude its applicability to problems where only the FW-oracle is efficiently solvable. We will show some examples that admit efficient away-oracle.

Before moving on to the analysis, we here make a new, albeit quick, observation that this selection scheme is in fact decomposing $\mathbf{x}_t$ implicitly. Specifically, it tries all possible decompositions of $\mathbf{x}_t$, and for each of them it finds the best away-direction in the traditional sense. Then it picks the best of the best over all proper convex decompositions of $\mathbf{x}_t$.

**Property 1.** *Denote $\mathcal{S}(\mathbf{x}) := \{S \subseteq \mathcal{P} : \mathbf{x} \text{ is a proper convex combination of all elements in } S\}$, where proper means that all elements in $S$ have a strictly positive weight. Then the away-step in* (3) *is exactly equivalent to $\max_{S \in \mathcal{S}(\mathbf{x}_t)} \max_{\mathbf{v} \in S} \langle \mathbf{v}, \nabla f(\mathbf{x}_t) \rangle$. See the proof in Appendix A.*

## 3 Analysis

We aim to analyze the rate by which the primal gap $h_t := f(\mathbf{x}_t) - f(\mathbf{x}^*)$ decays. Here $\mathbf{x}^*$ is the minimizer of $f$, and we assume it can be written as the convex combination of $s$ vertices of $\mathcal{P}$.

### 3.1 A New Geometric "Condition Number" of a Polytope

Underlying the analysis of linear convergence for FW-style algorithms is the following inequality that involves a geometric "condition number" $H_s$ of the polytope: ($\mathbf{v}_t^+$ and $\mathbf{v}_t^-$ are the FW and away-directions)

$$\sqrt{2H_s h_t/\alpha} \langle \mathbf{v}_t^+ - \mathbf{v}_t^-, \nabla f(\mathbf{x}_t) \rangle \le \langle \mathbf{x}^* - \mathbf{x}_t, \nabla f(\mathbf{x}_t) \rangle. \quad (4)$$

In Theorem 3 of [13], this $H_s$ is essentially the pyramidal width inverse. In Lemma 3 of [18], it is the cardinality of the optimal solution, which, despite being better than the pyramidal width, is restricted to SLPs. Our first key step here is to relax this restriction to arbitrary polytopes and define our $H_s$.

Let $\{\mathbf{u}_i\}$ be the set of vertices of the polytope $\mathcal{P}$, and this set must be finite. We do not assume $\mathbf{u}_i$ is binary. The following "margin" for each separating hyperplane directions $\mathbf{a}_k$ will be important:

$$g_k := \max_i \langle \mathbf{a}_k, \mathbf{u}_i \rangle - \text{second\_max}_i \langle \mathbf{a}_k, \mathbf{u}_i \rangle \ge 0. \quad (5)$$

Here the second max is the second *distinct* max in $\{\langle \mathbf{a}_k, \mathbf{u}_i \rangle : i\}$. If $\langle \mathbf{a}_k, \mathbf{u}_i \rangle$ is invariant to $i$, then this inequality $\langle \mathbf{a}_k, \mathbf{x} \rangle \le b_k$ is indeed an equality constraint ($\langle \mathbf{a}_k, \mathbf{x} \rangle = \max_{\mathbf{z} \in \mathcal{P}} \langle \mathbf{a}_k, \mathbf{z} \rangle$) hence can be moved to $C\mathbf{x} = \mathbf{d}$. So w.l.o.g, we assume $g_k > 0$. Now we state the generalized result.

**Lemma 1.** *Let $\mathcal{P}$ be defined as in* (1). *Suppose $\mathbf{x}$ can be written as some convex combination of $s$ number of vertices of $\mathcal{P}$: $\mathbf{x} = \sum_{i=1}^s \gamma_i \mathbf{u}_i$, where $\gamma_i \ge 0$, $\mathbf{1}^\top \boldsymbol{\gamma} = 1$. Then any $\mathbf{y} \in \mathcal{P}$ can be written as $\mathbf{y} = \sum_{i=1}^s (\gamma_i - \Delta_i)\mathbf{u}_i + (\mathbf{1}^\top \Delta)\mathbf{z}$, such that $\mathbf{z} \in \mathcal{P}$, $\Delta_i \in [0, \gamma_i]$, and $\mathbf{1}^\top \Delta \le \sqrt{H_s} \|\mathbf{x} - \mathbf{y}\|$ where*

$$H_s := \max_{S \subseteq [m], |S| = s} \sum_{j=1}^n \left( \sum_{k \in S} \frac{a_{kj}}{g_k} \right)^2. \quad (6)$$

*In addition, Equation* (4) *holds with this definition of $H_s$. Note our $H_s$ is defined here, not in* (4).

Some intuitive interpretations of $H_s$ are in order. First the definition in (6) admits a much more explicit characterization than pyramidal width. The maximization in (6) ranges over all possible subsets of constraints with cardinality $s$, and can hence be much lower than if $s = m$ (taking all constraints). Recall that pyramidal width is oblivious to, hence not benefiting from, the sparsity of the optimal solution. More comparisons are hard to make because [13] only provided an existential proof of pyramidal width, along with its value for simplex and hypercube only.[2]

However, $H_s$ is clearly *not* intrinsic of the polytope. For example, by definition $H_s = n$ for $\mathcal{Q}_2$. By contrast, we can introduce a slack variable $y$ to $\mathcal{Q}_2$, leading to a polytope over $[\mathbf{x}; y]$ (vertical

concatenation), with $\mathbf{x} \geq \mathbf{0}, y \geq 0, y + \mathbf{1}^\top \mathbf{x} = 2$. The augmented polytope enjoys $H_s = s$. Nevertheless, adding slack variables increases the diameter of the space and the vertices may no longer be binary. It also incurs more computation.

Second, $g_k$ may approach 0 (tending $H_s$ to infinity) when more linear constraints are introduced and vertices get closer neighbors. $H_s$ is infinity if the domain is not a polytope, requiring an uncountable number of supporting hyperplanes. Third, due to the square in (6), $H_s$ grows more rapidly as one variable participates in a larger number of constraints, than as a constraint involves a larger number of variables. When all $g_k = 1$ and all $a_{kj}$ are nonnegative, $H_s$ grows with the magnitude of $a_{kj}$. However this is not necessarily the case when $a_{kj}$ elements have mixed sign. Finally, $H_s$ is relative to the affine subspace that $P$ lies in, and is independent of linear equality constraints.

The proof of Lemma 1 utlizes the fact that the lowest value of $\mathbf{1}^\top \Delta$ is the optimal objective value of

$$\min_{\Delta, \mathbf{z}} \ \mathbf{1}^\top \Delta, \quad s.t. \quad \mathbf{0} \leq \Delta \leq \gamma, \quad \mathbf{y} = \mathbf{x} - (\mathbf{u}_1, \ldots, \mathbf{u}_s)\Delta + (\mathbf{1}^\top \Delta)\mathbf{z}, \quad \mathbf{z} \in \mathcal{P}, \quad (7)$$

where the inequalities are both elementwise. To ensure $\mathbf{z} \in \mathcal{P}$, we require $A\mathbf{z} \leq \mathbf{b}$, i.e.

$$(\mathbf{b}\mathbf{1}^\top - AU)\Delta \geq A(\mathbf{y} - \mathbf{x}), \quad \text{where} \quad U = (\mathbf{u}_1, \ldots, \mathbf{u}_s). \quad (8)$$

The rest of the proof utilizes the optimality conditions of $\Delta$, and is relegated to Appendix A. Compared with Lemma 2 of [18], our Lemma 1 does not require $\text{ext}(\mathcal{P})$ to be binary, and allows arbitrary inequality constraints rather than only $\mathbf{x} \geq \mathbf{0}$. Note $H_s$ depends on $\mathbf{b}$ indirectly, and employs a more explicit form for computation than pyramidal width. Obviously $H_s$ is non-decreasing in $s$.

**Example 1.** To get some idea, consider the $k$-simplex $\mathcal{P}_k$ or more general polytopes $\{\mathbf{x} \in [0, 1]^n : C\mathbf{x} = \mathbf{d}\}$. In this case, the inequality constraints are exclusively $x_i \in [0, 1]$, meaning $\mathbf{a}_k = \pm \mathbf{e}_k$ for all $k \in [2n]$ in (1). Here $\mathbf{e}_k$ stands for a canonical vector of straight 0 except a single 1 in the $k$-th coordinate. Obviously all $g_k = 1$. Therefore by Lemma 1, one can derive $H_s = s, \forall s \leq n$.

**Example 2.** To include inequality, let us consider $\mathcal{Q}_k$, the convex hull of a $k$-simplex. Lemma 1 implies its $H_s = n + 3s - 3$, independent of $k$. One might hope to get better $H_s$ when $k = 1$, since the constraint $\mathbf{x} \leq \mathbf{1}$ can be dropped in this case. Unfortunately, still $H_s = n$.

**Remark 1.** *The $L_0$ norm of the optimal $\mathbf{x}$ can be connected with $s$ simply by Caratheodory's theorem. Obviously $s = \|\mathbf{x}\|_0$ ($L_0$ norm) for $\mathcal{P}_1$ and $\mathcal{Q}_1$. In general, an $\mathbf{x}$ in $\mathcal{P}$ may be decomposed in multiple ways, and Lemma 1 immediately applies to the lowest (best) possible value of $s$ (which we will refer to as the **cardinality** of $\mathbf{x}$ following [18]). For example, the smallest $s$ for any $\mathbf{x} \in \mathcal{P}_k$ (or $\mathcal{Q}_k$) must be at most $\|\mathbf{x}\|_0 + 1$, because $\mathbf{x}$ must be in the convex hull of $V := \{\mathbf{y} \in \{0, 1\}^n : \mathbf{1}^\top \mathbf{y} = k, x_i = 0 \Rightarrow y_i = 0 \forall i\}$. Clearly its affine hull has dimension $\|\mathbf{x}\|_0$, and $V$ is a subset of $\text{ext}(\mathcal{P}_k) = \text{ext}(\mathcal{Q}_k)$.*

### 3.2 Tightness of $H_s$ under a Given Representation of the Polytope

We show some important examples that demonstrate the tightness of Lemma 1 with respect to the dimensionality ($n$) and the cardinality of $\mathbf{x}$ ($s$). Note the tightness is in the sense of satisfying the conditions in Lemma 1, not in the rate of convergence for the optimization algorithm.

**Example 3.** Consider $\mathcal{Q}_2$. $\mathbf{u}_1 = \mathbf{e}_1$ is a vertex and let $\mathbf{x} = \mathbf{u}_1$ (hence $s = 1$) and $\mathbf{y} = (1, \epsilon, \ldots, \epsilon)^\top$, where $\epsilon > 0$ is a small scalar. So in the necessary condition (8), the row corresponding to $\mathbf{1}^\top \mathbf{x} \leq 2$ becomes $\Delta_1 \geq (n-1)\epsilon = \sqrt{n-1} \cdot \|\mathbf{x} - \mathbf{y}\|$. By Lemma 1, $H_s = n$ which is almost $n - 1$.

**Example 4.** Let us see another example that is not simplex-like. Let $\mathbf{a}_k = -\mathbf{e}_k + \mathbf{e}_{n+1} + \mathbf{e}_{n+2}$ for $k \in [n]$. Let $A = (\mathbf{a}_1, \ldots, \mathbf{a}_n)^\top = (-I, \mathbf{1}, \mathbf{1})$ where $I$ is the identity matrix. Define $\mathcal{P}$ as $\mathcal{P} = \{\mathbf{x} \in [0, 1]^{n+2} : A\mathbf{x} \leq \mathbf{1}\}$, i.e. $\mathbf{b} = \mathbf{1}$. Since $A$ is totally unimodular, all the vertices of $\mathcal{P}$ must be binary. Let us consider $\mathbf{x} = \sum_{i=1}^n i\epsilon \mathbf{e}_i + r\mathbf{e}_{n+1} + (1 - r\epsilon)\mathbf{e}_{n+2}$, where $r = n(n+1)/2$ and $\epsilon > 0$ is a small positive constant. $\mathbf{x}$ can be represented as the convex combination of $n + 1$ vertices

$$\mathbf{x} = \sum_{i=1}^n i\epsilon \mathbf{u}_i + (1 - r\epsilon)\mathbf{u}_{n+1}, \quad \text{where} \quad \mathbf{u}_i = \mathbf{e}_i + \mathbf{e}_{n+1} \text{ for } i \leq n, \text{ and } \mathbf{u}_{n+1} = \mathbf{e}_{n+2}. \quad (9)$$

With $U = (\mathbf{u}_1, \ldots, \mathbf{u}_{n+1})$, we have $\mathbf{b}\mathbf{1}^\top - AU = (I, \mathbf{0})$. Let $\mathbf{y} = \mathbf{x} + \epsilon \mathbf{e}_{n+1}$, which is clearly in $\mathcal{P}$. Then (8) becomes $\Delta \geq \epsilon \mathbf{1}$, and so $\mathbf{1}^\top \Delta \geq \sqrt{n^2} \|\mathbf{y} - \mathbf{x}\|$. Applying Lemma 1 with $s = n + 1$ and $g_k = 1$ for all $k$, we get $H_s = 2n^2 + n - 1$, which is of the same order of magnitude as $n^2$.

### 3.3 Analysis for Pairwise Frank-Wolfe (PFW-1) on SLPs

Equipped with Lemma 1, we can now extend the analysis in [18] to SLPs where the constraint of $\mathbf{x} \leq \mathbf{1}$ can be explicitly accommodated *without having to introduce a slack variable* which increases the diameter $D$ and costs more computations.

**Theorem 1.** *Applying PFW-1 to SLP, all iterates must be feasible and $h_t \leq \frac{\beta D^2}{2}(1-c_1)^{t-1}$ if we set $\gamma_t = c_1^{1/2}(1-c_1)^{\frac{t-1}{2}}$, where $c_1 = \frac{\alpha}{16\beta H_s D^2}$. The proof just replaces all $card(\mathbf{x}^*)$ in [18] with $H_s$.*

Slight effort is needed to guarantee the feasibility and we show it as Lemma 6 in Appendix A.

When $\mathcal{P}$ is not an SLP or general inequality constraints are present, we resort to line search (PFW-2), which is more efficient than PFW-1 in practice. However, the analysis becomes challenging [13, 18], because it is difficult to bound the number of steps where the step size is clamped due to the feasibility constraint (the swap step in [13]). So [13] appealed to a bound that is the *factorial* of the number of vertices. Fortunately, we will show below that by switching to AFW, the line search version achieves linear rates with improved dimension dependency for general polytopes, and the pre-defined step version preserves the strong rates of PFW-1 on SLPs. These are all facilitated by the $H_s$ in Lemma 1.

### 3.4 Analysis for Away-step Frank-Wolfe with Pre-defined Step Size (AFW-1) on SLPs

We first show that AFW-1 achieves the same rate of convergence as PFW-1 on SLPs. Although this does not appear surprising and the proof architecture is similar to [18], we stress that the step size needs delicate modifications because the descent direction $\mathbf{d}_t$ in PFW does not rescale $\mathbf{x}_t$, while AFW does. Our key novelty is to first run a constant number of FW-steps ($O(\frac{1}{t})$ rate), and start accepting away-steps when the step size is small enough to ensure feasibility and linear convergence.

We first establish the feasibility of iterates under the pre-defined step sizes. Proofs are in Appendix A.

**Lemma 2** (Feasibility of iterates for AFW-1). *Suppose $\mathcal{P}$ is an SLP and the reference step sizes $\{\gamma_t\}_{t\geq n_0}$ are contained in $[0, 1]$. Then the iterates generated by AFW-1 are always feasible.*

**Choosing the step size.** Key to the AFW-1 algorithm is the *delicately* chosen sequence of step sizes. For AFW-1, define (logarithms are natural basis)

$$\gamma_t = \frac{M_1}{\theta M_2}\sqrt{c_0}(1-c_1)^{(t-1)/2}, \quad \text{where} \quad M_1 = \sqrt{\frac{\alpha}{8H_s}}, \quad M_2 = \frac{\beta D^2}{2}, \quad \theta = 52 \qquad (10)$$

$$c_1 = \frac{M_1^2}{M_2}\frac{\theta-4}{4\theta^2} < \frac{1}{200}, \quad n_0 = \left\lceil \frac{1}{c_1} \right\rceil, \quad c_0 = \frac{3M_2 \log n_0}{n_0}(1-c_1)^{1-n_0}. \qquad (11)$$

**Lemma 3.** *In AFW-1, we have $h_t \leq \frac{3}{t}M_2 \log t$ for all $t \in [2, n_0]$. Obviously $n_0 \geq 200$ by (11).*

This result is similar to Theorem 1 in [4]. However, their step size is $2/(t+2)$ leading to a $\frac{2}{t+2}M_2$ rate of convergence. Such a step size will break the integrality of the iterates, and hence we adjusted the step size, at the cost of a $\log t$ term in the rates which can be easily handled in the sequel.

The condition number $c_1$ gets better (bigger) when: the strongly convex parameter $\alpha$ is larger, the smoothness constant $\beta$ is smaller, the diameter D of the domain is smaller, and $H_s$ is smaller.

**Lemma 4.** *For all $t \geq n_0$, AFW-1 satisfies a) $\gamma_t \leq 1$, b) $\gamma_{t+1}^{-1} - \gamma_t^{-1} \geq 1$, and c) $\eta_t \in [\frac{1}{4}\gamma_t, \gamma_t]$.*

By Lemma 2 and Lemma 4a, we know that the iterates generated by AFW-1 are all feasible.

**Theorem 2.** *Applying AFW-1 to SLP, the gap decays as $h_t \leq c_0(1-c_1)^{t-1}$ for all $t \geq n_0$.*

*Proof.* By Lemma 3, $h_{n_0} \leq \frac{3M_2}{n_0}\log n_0 = c_0(1-c_1)^{n_0-1}$. Let the result hold for some $t \geq n_0$. Then

$$h_{t+1} \leq h_t + \eta_t \langle \mathbf{d}_t, \nabla f(\mathbf{x}_t)\rangle + \frac{\beta}{2}\eta_t^2 D^2 \quad \text{(smoothness of } f) \tag{12}$$

$$\leq h_t + \frac{\eta_t}{2}\langle \mathbf{v}_t^+ - \mathbf{v}_t^-, \nabla f(\mathbf{x}_t)\rangle + \frac{\beta}{2}\eta_t^2 D^2 \quad \text{(by step 5 of Algorithm 1)} \tag{13}$$

$$\leq h_t - \frac{\eta_t}{2}\sqrt{\frac{\alpha}{2H_s}}\sqrt{h_t} + \frac{\beta}{2}\eta_t^2 D^2 \quad \text{(by (4) and the fact } \langle \mathbf{x}^* - \mathbf{x}_t, \nabla f(\mathbf{x}_t)\rangle \leq -h_t) \tag{14}$$

$$\leq h_t - \frac{1}{4}M_1\gamma_t h_t^{1/2} + \frac{\beta}{2}\gamma_t^2 D^2 \quad \text{(Lemma 4c and the defn. of } M_1) \tag{15}$$

$$= h_t - \frac{M_1^2}{4\theta M_2}\sqrt{c_0}(1-c_1)^{(t-1)/2}h_t^{1/2} + \frac{M_1^2}{\theta^2 M_2}c_0(1-c_1)^{t-1} \quad \text{(by defn. of } \gamma_t) \tag{16}$$

$$\leq c_0(1-c_1)^{t-1}\left(1 - \frac{M_1^2}{4\theta M_2} + \frac{M_1^2}{\theta^2 M_2}\right) = c_0(1-c_1)^t \quad \text{(by defn. of } c_1). \tag{17}$$

Here the inequality in step (17) is by treating (16) as a quadratic of $h_t^{1/2}$ and applying the induction assumption on $h_t$. The last step completes the induction: the conclusion also holds for step $t+1$. $\square$

### 3.5 Analysis for Away-step Frank-Wolfe with Line Search (AFW-2)

We finally analyze AFW-2 on general polytopes with line search. Noting that $f(\mathbf{x}_t + \eta \mathbf{d}_t) - f(\mathbf{x}^*) \leq$ (14) (with $\eta_t$ in (14) replaced by $\eta$), we minimize both sides over $\eta : \mathbf{x}_t + \eta \mathbf{d}_t \in \mathcal{P}$. If none of the inequality constraints are satisfied as equality at the optimal $\eta_t$ of line search, then we call it a *good step* and in this case

$$h_{t+1} \leq \left(1 - \frac{\alpha}{256\beta D^2 H_s}\right) h_t, \qquad \text{(Eq 14 in } \eta \text{ is minimized at } \eta_t^* := \frac{1}{\beta D^2} M_1 h_t^{1/2}). \qquad (18)$$

The only task left is to bound the number of bad steps (*i.e.* $\eta_t$ clamped by its upper bound). In [13] where the set of atoms is maintained, it is easily shown that up to step $t$ there can be only at most $t/2$ bad steps, and so the overall rate of convergence is slowed down by at most a factor of two. This favorable result no longer holds in our decomposition-invariant AFW. However, thanks to the special property of AFW, it is still not hard to bound the number of bad steps between two good steps.

First we notice that such clamping never happens for FW-steps, because $\eta_t^* \leq 1$ and for FW-steps, $\mathbf{x}_t + \eta_t \mathbf{d}_t \in \mathcal{P}$ implicitly enforces $\eta_t \leq 1$ only (after $\eta_t \geq 0$ is imposed). For an away-step, if the line search is blocked by some constraint, then at least one inequality constraint will turn into an equality constraint if the next step is still away. Since AFW selects the away-direction by respecting all equality constraints, the succession of away-steps (called an *away epoch*) must terminate when the set of equalities define a *singleton*. For any index set of inequality constraints $S \subseteq [m]$, let $\mathcal{P}(S) := \{\mathbf{x} \in \mathcal{P} : \langle \mathbf{a}_j, \mathbf{x} \rangle = b_j, \forall j \in S\}$ be the set of points that satisfy these inequalities with equality. Let

$$n(\mathcal{P}) := \max \{|S| : S \subseteq [m], |\mathcal{P}(S)| = 1, |\mathcal{P}(S')| = \infty \text{ for all } S' \subsetneq S\} \qquad (19)$$

be the maxi-min number of constraints to define a singleton. Then obviously $n(\mathcal{P}) \leq n$, and so

**Theorem 3.** *To find an $\epsilon$ accurate solution, AFW-2 requires at most* $O\left(\frac{n\beta D^2 H_s}{\alpha} \log \frac{1}{\epsilon}\right)$ *steps.*

**Example 5.** Suppose $f(\mathbf{x}) = \frac{1}{2}\|\mathbf{x} + \mathbf{1}\|^2$ with $\mathcal{P} = [0,1]^n$. Clearly $n(\mathcal{P}) = n$. Unfortunately we can construct an initial $\mathbf{x}_1$ as a convex combination of only $O(\log n)$ vertices, but AFW-2 will then run $O(n)$ number of away-steps consecutively. Hence our above analysis on the max length of away epoch seems tight, although having $n$ consecutive away-steps between two good steps *once* is different than this happening multiple times. See the construction of $\mathbf{x}_1$ in Appendix A.

**Tighter bounds.** By refining the analysis of the polytopes, we may improve upon the $n(\mathcal{P})$ bound. For example it is not hard to show that $n(\mathcal{P}_k) = n(\mathcal{Q}_k) = n$. Let us consider the number of non-zeros in the iterates $\mathbf{x}_t$. A bad step (which must be an away-step) will either a) set an entry to 1, which will force the corresponding entry of $\mathbf{v}_t^-$ to be 1 in the future steps of the away epoch, hence can happen at most $k$ times; or b) set at least one nonzero entry of $\mathbf{x}_t$ into 0, and will never switch a zero entry to nonzero. But each FW-step may introduce at most $k$ nonzeros. So the number of bad steps cannot be over $2k$ times of that of FW-step, and the overall iteration complexity is at most $O(\frac{k\beta D^2 H_s}{\alpha} \log \frac{1}{\epsilon})$.

We can now revisit Table 1 and observe the generality and efficiency of AFW-2. It is noteworthy that on SLPs, we are not yet able to establish the same rate as AFW-1. We believe that the vertices being binary is very special, making it hard to generalize the analysis.

## 4 Application to Kernel Binary SVM

As a concrete example, we apply AFW to the dual objective of a binary SVM with bias:

$$\text{(SVM-Dual)} \qquad \min_{\mathbf{x}} f(\mathbf{x}) := \frac{1}{2}\mathbf{x}^\top Q \mathbf{x} - \frac{1}{C}\mathbf{1}^\top \mathbf{x}, \quad s.t. \quad \mathbf{x} \in [0,1]^n, \quad \mathbf{y}^\top \mathbf{x} = 0. \qquad (20)$$

Here $\mathbf{y} = (y_1, \ldots, y_n)^\top$ is the label vector with $y_i \in \{-1, 1\}$, and $Q$ is the signed kernel matrix with $Q_{ij} = y_i y_j k(\mathbf{x}_i, \mathbf{x}_j)$. Since the feasible region is an SLP with diameter $O(\sqrt{n})$, we can use both AFW-1 and PFW-1 to solve it with $O(\#\text{SV} \cdot n\kappa \log \frac{1}{\epsilon})$ iterations, where $\kappa$ is the ratio between the maximum and minimum eigenvalues of $Q$ (assume $Q$ is positive definite), and #SV stands for the number of support vectors in the optimal solution.

**Computational efficiency per iteration.** The key technique for computational efficiency is to keep updating the gradient $\nabla f(\mathbf{x})$ over the iterations, exploiting the fact that $\mathbf{v}_t^+$ and $\mathbf{v}_t^-$ might be sparse and $\nabla f(\mathbf{x}) = Q\mathbf{x} - \frac{1}{C}\mathbf{1}$ is affine in $\mathbf{x}$. In particular, when AFW takes a FW-step in line 5, we have

$$Q\mathbf{d}_t = Q\mathbf{d}_t^{\mathrm{FW}} = Q(\mathbf{v}_t^+ - \mathbf{x}_t) = -\nabla f(\mathbf{x}_t) - \frac{1}{C}\mathbf{1} + Q\mathbf{v}_t^+. \tag{21}$$

Similar update formulas can be shown for away-step $\mathbf{d}_t^A$ and PFW-step $\mathbf{d}_t^{\mathrm{PFW}}$. So if $\mathbf{v}^+$ (or $\mathbf{v}_t^-$) has $k$ non-zeros, all these three updates can be performed in $O(kn)$ time. Based on them, we can update the gradient by $\nabla f(\mathbf{x}_{t+1}) = \nabla f(\mathbf{x}_t) + \eta_t Q\mathbf{d}_t$. The FW-oracle and away-oracle cost $O(n)$ time given the gradient, and the line search has a closed form solution. See more details in Appendix B.

**Major drawback.** This approach unfortunately provides no control of the sparseness of $\mathbf{v}_t^+$ and $\mathbf{v}_t^-$. As a result, each iteration may require evaluating the entire kernel matrix ($O(n^2)$ kernel evaluations), leading to an overall computational cost $O(\#\mathrm{SV} \cdot n^3 \kappa \log \frac{1}{\epsilon})$. This can be prohibitive.

## 4.1 Reformulation by Relaxed Convex Hull

To ensure the sparsity of each update, we reformulate the SVM dual objective (20) by using the reduced convex hull (RC-Hull, [22]). Let $P$ and $N$ be the set of positive and negative examples, resp.

(RC-Margin) $\displaystyle \min_{\boldsymbol{\theta}, \, \boldsymbol{\xi}^+ \in \mathbb{R}^{|P|}, \, \boldsymbol{\xi}^- \in \mathbb{R}^{|N|}, \, \alpha, \, \beta} \frac{1}{K}(\mathbf{1}^\top \boldsymbol{\xi}^+ + \mathbf{1}^\top \boldsymbol{\xi}^-) + \frac{1}{2}\|\boldsymbol{\theta}\|^2 - \alpha + \beta,$

$$s.t. \quad A^\top \boldsymbol{\theta} - \alpha\mathbf{1} + \boldsymbol{\xi}^+ \geq \mathbf{0}, \quad -B^\top \boldsymbol{\theta} + \beta\mathbf{1} + \boldsymbol{\xi}^- \geq \mathbf{0}, \quad \boldsymbol{\xi}^+ \geq \mathbf{0}, \quad \boldsymbol{\xi}^- \geq \mathbf{0}. \tag{22}$$

(RC-Hull) $\displaystyle \min_{\mathbf{u}\in\mathbb{R}^{|P|}, \mathbf{v}\in\mathbb{R}^{|N|}} \frac{1}{2}\|A\mathbf{u} - B\mathbf{v}\|^2, \quad s.t. \quad \mathbf{u} \in \mathcal{P}_K, \, \mathbf{v} \in \mathcal{P}_K. \tag{23}$

Here $A$ (or $B$) is a matrix whose $i$-th column is the (implicit) feature representation of the $i$-th positive (or negative) example. RC-Margin resembles the primal SVM formulation, except that the bias term is split into two terms $\alpha$ and $\beta$. RC-Hull is the dual problem of RC-Margin, and it has a very intuitive geometric meaning. When $K = 1$, RC-Hull tries to find the distance between the convex hull of $P$ and $N$. When the integer $K$ is greater than 1, then $\frac{1}{K}A\mathbf{u}$ is a *reduced convex hull* of the positive examples, and the objective finds the distance of the reduced convex hull of $P$ and $N$.

Since the feasible region of RC-Hull is a simplex, $\mathbf{d}_t$ in AFW and PFW have at most $2K$ and $4K$ nonzeros respectively, and it costs $O(nK)$ time to update the gradient (see Appendix B.1). Given $K$, Appendix B.2 shows how to recover the corresponding $C$ in (20), and to translate the optimal solutions. Although solving RC-Hull requires the knowledge of $K$ (which is unknown a priori if we are only given $C$), in practice, it is equally justified to tune the value of $K$ via model selection tools in the first place, which is approximately tuning the number of support vectors.

## 4.2 Discussion and Comparison of Rates of Convergence

Clearly, the feasible region of RC-Hull is an SLP, allowing us to apply AFW-1 and PFW-1 with optimal linear convergence: $O(\#\mathrm{SV} \cdot \kappa K \log \frac{1}{\epsilon}) \leq O(\kappa(\#\mathrm{SV})^2 \log \frac{1}{\epsilon})$, because $K = \mathbf{1}^\top \mathbf{u} \leq \#\mathrm{SV}$. So overall, the computational cost is $O(n\kappa(\#\mathrm{SV})^3 \log \frac{1}{\epsilon})$.

[20] shows sequential minimal optimization (SMO) [19, 23] costs $O(n^3 \kappa \log \frac{1}{\epsilon})$ computations. This is greater than $O(n\kappa(\#\mathrm{SV})^3 \log \frac{1}{\epsilon})$ when $\#\mathrm{SV} \leq n^{2/3}$. [24] requires $O(\kappa^2 n \|Q\|_{\mathrm{sp}} \log \frac{1}{\epsilon})$ iterations, and each iteration costs $O(n)$. SVRG [25], SAGA [26], SDCA [27] require losses to be decomposable and smooth, which do not hold for hinge loss with a bias. SDCA can be extended to almost smooth losses such as hinge loss, but still the dimension dependency is unclear and it cannot handle bias.

As a final remark, despite the superior rates of AFW-1 and PFW-1, their pre-defined step size makes them impractical. With line search, AFW-2 is much more efficient in practice, *and* at the same time provides theoretical guarantees of $O(n\kappa(\#\mathrm{SV})^4 \log \frac{1}{\epsilon})$ computational cost, just slightly worse by $\#\mathrm{SV}$ times. Such an advantage in *both* theory and practice *by one method* is not available in PFW [18].

# 5 Experiments and Future Work

In this section we compare the empirical performance of AFW-2 against related methods. We first illustrate the performance on kernel binary SVM, then we investigate a problem whose domain is not an SLP, and finally we demonstrate the scalability of AFW-2 on a large scale dataset.

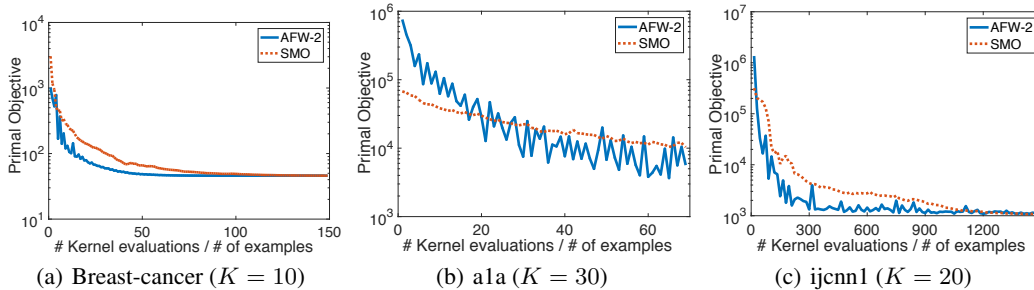

(a) Breast-cancer ($K = 10$)      (b) a1a ($K = 30$)      (c) ijcnn1 ($K = 20$)

Figure 1: Comparison of SMO and AFW-2 on three different datasets

**Binary SVM** Our first comparison is on solving kernel binary SVMs with bias. Three datasets are used. *breast-cancer* and *a1a* are obtained from the UCI repository [28] with $n = 568$ and $1,605$ training examples respectively, and *ijcnn1* is from [29] with a subset of $5,000$ examples.

As a competitor, we adopted the well established Sequential Minimal Optimization (SMO) algorithm [19]. The implementation updates all cached errors corresponding to each examples if any variable is being updated at each step. Using these cached error, the algorithm heuristically picks the best subset of variable to update at each iteration.

We first run AFW-2 on the RC-Hull objective in (23), with the value of $K$ set to optimize the test accuracy ($K$ shown in Figure 1). After obtaining the optimal solution, we compute the equivalent $C$ value based on the conversion rule in Appendix B.2, and then run SMO on the dual objective (20).

Figure 1 shows the decay of the primal SVM objective (hence fluctuation) as a function of (the number of kernel evaluations divided by $n$). This avoids the complication of CPU frequency and kernel caching. Clearly, AFW-2 outperforms SMO on *breast-cancer* and *ijcnn1*, and overtakes SMO on *a1a* after a few iterations.

PFW-1 and PFW-2 are also applicable to the RC-Hull formulation. Although the rate of PFW-1 is better than AFW-2, it is much slower in practice. Although empirically we observed that PFW-2 is similar to our AFW-2, unfortunately PFW-2 has no theoretical guarantee.

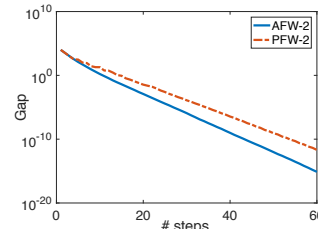

Figure 2: Least square w. $\mathcal{Q}_{375}$

**General Polytope** Our next comparison uses $\mathcal{Q}_k$ as the domain. Since it is not an SLP, neither PFW-1 nor PFW-2 provides a bound. Here we aim to show that AFP-2 is not only advantageous in providing a good rate of convergence, it is also comparable to (or better than) PFW-2 in terms of practical efficiency. Our objective is a least square (akin to lasso):

$$\min_{\mathbf{x}} f(\mathbf{x}) = \|A\mathbf{x} - \mathbf{b}\|^2, \quad \mathbf{0} \leq \mathbf{x} \leq \mathbf{1}, \quad \mathbf{1}^{\top}\mathbf{x} \leq 375.$$

Here $A \in \mathbb{R}^{100 \times 1000}$, and both $A$ and $\mathbf{b}$ were generated randomly. Both the FW-oracle and away-oracle are simply based on sorting the gradient. As shown in Figure 2, AFW-2 is indeed slightly faster than PFW-2.

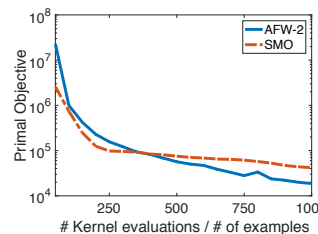

Figure 3: Full ijcnn1

**Scalability** To demonstrate the scalability of AFP-2, we plot its convergence curve ($K = 100$) along with SMO on the full *ijcnn1* dataset with $49,990$ examples. In Figure 3, AFW-2 starts with a higher primal objective value, but after a while it outperforms SMO near the optimum. In this problem, kernel evaluation is the major computational bottleneck, hence used as the horizontal axis. This also helps avoiding the complication of CPU speed (*e.g.* when wall-clock time is used).

# 6 Future work

We will extend the decomposition invariant method to gauge regularized problems [30–32], and derive comparable linear convergence rates. Moreover, although it is hard to evaluate pyramidal width, it will be valuable to compare it with $H_s$ even in terms of upper/lower bounds.

**Acknowledgements.** We thank Dan Garber for very helpful discussions and clarifications on [18]. Mohammad Ali Bashiri is supported in part by NSF grant RI-1526379.

## Footnotes

[1]Although [18] does not allow for $\mathbf{x} \leq \mathbf{1}$ constraints, we can add a slack variable $y_i$: $y_i + x_i = 1$, $y_i \geq 0$.

[2][21] showed pyramidal width is equivalent to a more interpretable quantity called "facial distance", and they derived its value for more examples. But the evaluation of its value remains challenging in general.

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
