[Supplementary Material]

# A Proofs

*Proof of Property 1.* Suppose $\mathbf{v} \in S \in \mathcal{S}(\mathbf{x}_t)$. Then there exists $\alpha \in (0,1)$ and $\mathbf{z} \in \mathcal{P}$ such that $\mathbf{x}_t = \alpha \mathbf{v} + (1-\alpha)\mathbf{z}$. If $\langle \mathbf{a}_j, \mathbf{x}_t \rangle = b_j$, then the fact that $\langle \mathbf{a}_j, \mathbf{z} \rangle \leq b_j$ implies that $\langle \mathbf{a}_j, \mathbf{v} \rangle = b_j$. Conversely, suppose $\mathbf{v} \in \mathcal{P}$ satisfies $\langle \mathbf{a}_j, \mathbf{x}_t \rangle = b_j \Rightarrow \langle \mathbf{a}_j, \mathbf{v} \rangle = b_j$ for all $j$. Then consider $\mathbf{z}_\alpha := \frac{1}{1-\alpha}(\mathbf{x}_t - \alpha \mathbf{v})$ for $\alpha \in (0,1)$. If $j$ satisfies $\langle \mathbf{a}_j, \mathbf{x}_t \rangle = b_j$, then clearly $\langle \mathbf{a}_j, \mathbf{z}_\alpha \rangle = b_j$. Otherwise if $\langle \mathbf{a}_j, \mathbf{x}_t \rangle < b_j$, then $\lim_{\alpha \downarrow 0} \langle \mathbf{a}_j, \mathbf{z}_\alpha \rangle = \langle \mathbf{a}_j, \mathbf{x}_t \rangle < b_j$. Since the number of inequality constraint is finite, we can guarantee that $\mathbf{z}_\alpha \in \mathcal{P}$ as long as the value of $\alpha$ is small enough. $\qquad\square$

*Proof of Lemma 1.* Denote $U = (\mathbf{u}_1, \ldots, \mathbf{u}_s)$. Clearly the lowest possible value of $\mathbf{1}^\top \Delta$ is at most the solution to the following optimization problem

$$\min_{\Delta, \mathbf{z}} \ \mathbf{1}^\top \Delta \tag{24}$$

$$s.t. \quad \mathbf{0} \leq \Delta \leq \gamma \tag{25}$$

$$\mathbf{y} = \mathbf{x} - U\Delta + (\mathbf{1}^\top \Delta)\mathbf{z} \tag{26}$$

$$\mathbf{z} \in \mathcal{P}, \tag{27}$$

where the inequalities are both elementwise. Obviously the feasible region is not empty because $\Delta = \gamma$ and $\mathbf{z} = \mathbf{y}$ is always feasible. When $\Delta = \mathbf{0}$ is feasible (*i.e.* $\mathbf{y} = \mathbf{x}$), (33) is obviously satisfied. Otherwise, we have

$$\mathbf{z} = (\mathbf{1}^\top \Delta)^{-1} (\mathbf{y} - \mathbf{x} + U\Delta) \in \mathcal{P}. \tag{28}$$

Notice that $C\mathbf{z} = \mathbf{d}$ is automatically satisfied because by $\mathbf{x}, \mathbf{y}, \mathbf{u}_i$ all lying in $\mathcal{P}$, we have

$$C\mathbf{z} = (\mathbf{1}^\top \Delta)^{-1} (C\mathbf{y} - C\mathbf{x} + CU\Delta) = (\mathbf{1}^\top \Delta)^{-1} (\mathbf{d} - \mathbf{d} + \mathbf{d}\mathbf{1}^\top \Delta) = \mathbf{d}. \tag{29}$$

So to ensure $\mathbf{z} \in \mathcal{P}$, we just need to further enforce $A\mathbf{z} \leq \mathbf{b}$, which is equivalent to:

$$(\mathbf{b}\mathbf{1}^\top - AU)\Delta \geq A(\mathbf{y} - \mathbf{x}). \tag{30}$$

Denote $F = \mathbf{b}\mathbf{1}^\top - AU$. Then by the definition of $g_k$, all entries in the $k$-th row of $F$ are either 0, or at least $g_k$. For any $i \in [s]$, there exists a row index $k_i$ of $F$ such that $F_{k_i, i} > 0$ and the inequality in (30) holds with equality for the $k_i$-th row. This is because, we can otherwise further reduce $\Delta_i$ to improve the objective function. Denoting by $I(k_i)$ the set of columns that are not zero in the $k_i$-th row of $F$, we now have

$$F_{k_i,:}\Delta = \mathbf{a}_{k_i}^\top (\mathbf{y} - \mathbf{x}) \quad \Rightarrow \quad \mathbf{a}_{k_i}^\top (\mathbf{y} - \mathbf{x}) \geq g_{k_i} \sum_{j \in I(k_i)} \Delta_j. \tag{31}$$

Therefore, denoting $K = \{k_i : i \in [s]\}$, we have $|K| \leq s$ and we finally arrive at

$$\sum_{i=1}^s \Delta_i \leq \sum_{k \in K} \sum_{i \in I(k)} \Delta_i \leq \sum_{k \in K} \frac{1}{g_k} \mathbf{a}_k^\top (\mathbf{y} - \mathbf{x}) = \sum_{j=1}^n \left[ \left( \sum_{k \in K} \frac{a_{kj}}{g_k} \right) (y_j - x_j) \right] \tag{32}$$

$$\leq \|\mathbf{y} - \mathbf{x}\| \left[ \sum_{j=1}^n \left( \sum_{k \in K} \frac{a_{kj}}{g_k} \right)^2 \right]^{1/2} \leq H_s \|\mathbf{y} - \mathbf{x}\|. \qquad\square$$

Incidentally, if $\mathcal{P}$ is not a polytope, then generally there is some $\mathbf{a}_k$ such that the $g_k$ defined in (5) is 0, even though $\mathbf{a}_k$ is not an equality constraint. Besides there can be an uncountable number of linear inequality constraints to define, say, a unit $L_2$ ball.

Before proving (4), we need a slight enhancement of Lemma 1 that swaps the role of $\mathbf{x}$ and $\mathbf{y}$.

**Lemma 5.** *Let $\mathbf{x}, \mathbf{y} \in \mathcal{P}$. Suppose $\mathbf{y}$ can be written as the convex hull of $s$ vertices of $\mathcal{P}$. Then we can write $\mathbf{x}$ as the convex combination of vertices of $\mathcal{P}$, $\mathbf{x} = \sum_{i=1}^k \lambda_i \mathbf{v}_i$ for some integer $k$, such that $\mathbf{y}$ can be written as $\mathbf{y} = \sum_{i=1}^k (\lambda_i - \Delta_i)\mathbf{v}_i + (\mathbf{1}^\top \Delta)\mathbf{z}$ with $\Delta_i \in [0, \lambda_i]$ for all $i \in [k]$, $\mathbf{z} \in \mathcal{P}$, and*

$$\mathbf{1}^\top \Delta \leq \sqrt{H_s} \|\mathbf{x} - \mathbf{y}\|. \tag{33}$$

*Proof of Lemma 5.* By assumption we can write $\mathbf{y} = \sum_{i=1}^{s} \gamma_i \mathbf{u}_i$ for $\mathbf{u}_i$ being vertices of $\mathcal{P}$, $\gamma_i \geq 0$, and $\mathbf{1}^\top \boldsymbol{\gamma} = 1$. By Lemma 1, $\mathbf{x}$ can be written as $\mathbf{x} = \sum_{i=1}^{s} (\gamma_i - \delta_i) \mathbf{u}_i + (\mathbf{1}^\top \delta) \mathbf{w}$, where $\mathbf{w} \in \mathcal{P}$, $\delta_i \in [0, \gamma_i]$, and $\sum_{i=1}^{s} \delta_i \leq H_s \|\mathbf{x} - \mathbf{y}\|$.

Now suppose $\mathbf{w} = \sum_{j=1}^{t} \alpha_j \mathbf{s}_j$, where $\mathbf{s}_j$ are vertices of $\mathcal{P}$, $\alpha_j \in [0, 1]$ and $\mathbf{1}^\top \boldsymbol{\alpha} = 1$. Letting $r = \mathbf{1}^\top \delta$, we have

$$\mathbf{x} = \sum_i \underbrace{(\gamma_i - \delta_i)}_{\lambda_i} \mathbf{u}_i + \sum_j \underbrace{(r\alpha_j)}_{\lambda_j'} \mathbf{s}_j \tag{34}$$

$$\mathbf{y} = \sum_i \underbrace{(\gamma_i - \delta_i}_{\lambda_i} - \underbrace{0}_{\Delta_i}) \mathbf{u}_i + \sum_j (\underbrace{r\alpha_j}_{\lambda_j'} - \underbrace{r\alpha_j}_{\Delta_j'}) \mathbf{s}_j + r \underbrace{\sum_i \frac{\delta_i}{r} \mathbf{u}_i}_{\mathbf{z}}. \tag{35}$$

So now we have found a decomposition of $\mathbf{x}$, where $\{\mathbf{v}_i\}$ corresponds to the union of $\{\mathbf{u}_i\}$ (with weights $\lambda_i = \gamma_i - \delta_i$) and $\{\mathbf{s}_j\}$ (with weights $\lambda_j' = r\alpha_j$). Furthermore, $\Delta_i = 0$ for $\mathbf{u}_i$ and $\Delta_j' = r\alpha_j$ for $\mathbf{s}_j$. $\mathbf{z}$ corresponds to $\sum_i \frac{\delta_i}{r} \mathbf{u}_i \in \mathcal{P}$, and notice that $\sum_i \Delta_i + \sum_j \Delta_j' = r = \sum_i \delta_i$. $\qquad\square$

One might thus wonder why we do not eliminate inequality constraints altogether by introducing slack variables. The answer is that first the diameter $D$ of the new polytope will grow in the number of constraints which can be arbitrarily higher than the original dimensionality, and the rate of convergence depends on $D^2$. Second, even if the original polytope has all vertices being binary, the vertices of the augmented polytope are not necessarily binary (*e.g.* $\mathcal{Q}_k$ with $y = k - \mathbf{1}^\top \mathbf{x}$). So in the sequel, we will not complicate ourselves with "smart" reformulations of the polytope.

*Proof of Equation* (4). By strong convexity, we have $\sqrt{2 H_s h_t / \alpha} \geq \sqrt{H_s} \|\mathbf{x}_t - \mathbf{x}^*\|$. By Lemma 5, we can write $\mathbf{x}_t$ as a convex combination of $\mathbf{x}_t = \sum_{i=1}^{k} \mathbf{u}_i$ and $\mathbf{x}^*$ as $\mathbf{x}^* = \sum_{i=1}^{k} (\lambda_i - \Delta_i) \mathbf{v}_i + (\mathbf{1}^\top \Delta) \mathbf{z}$, where $\Delta_i \in [0, \lambda_i]$, $\mathbf{z} \in \mathcal{P}$, and $\mathbf{1}^\top \Delta \leq \sqrt{H_s} \|\mathbf{x}_t - \mathbf{x}^*\| \leq \sqrt{2 H_s h_t / \alpha}$. Therefore, we get the first inequality in (4) by

$$\left\langle \sqrt{2 H_s h_t / \alpha} (\mathbf{v}_t^+ - \mathbf{v}_t^-), \nabla f(\mathbf{x}_t) \right\rangle \leq \sum_{i=1}^{k} \Delta_i \left\langle \mathbf{v}_t^+ - \mathbf{v}_t^-, \nabla f(\mathbf{x}_t) \right\rangle \tag{36}$$

$$\leq \sum_{i=1}^{k} \Delta_i \left\langle \mathbf{z} - \mathbf{v}_i, \nabla f(\mathbf{x}_t) \right\rangle = \left\langle \mathbf{x}^* - \mathbf{x}_t, \nabla f(\mathbf{x}_t) \right\rangle, \tag{37}$$

where the first inequality follows since $\left\langle \mathbf{v}_t^+ - \mathbf{v}_t^-, \nabla f(\mathbf{x}_t) \right\rangle \leq 0$, and the second inequality follows from the optimality of $\mathbf{v}_t^+$ and $\mathbf{v}_t^-$ (Property 1). $\qquad\square$

**Lemma 6** (Feasibility of iterates for PFW-1). *Suppose $\mathcal{P}$ is an SLP, and the reference step sizes $\{\gamma_t\}_{t \geq 1}$ are contained in $[0, 1]$. Then the iterates generated by PFW-1 are always feasible.*

*Proof of Lemma 6.* We prove by induction that $\mathbf{s}_t := \mathbf{x}_t / \eta_t = q_t \mathbf{x}_t$ is integral in all coordinates and $\mathbf{x}_t \in [0, 1]^n$. When $t = 1$, since $\mathbf{x}_1$ is an extreme point, it must lie in $\{0, 1\}^n$. Then $\mathbf{s}_1 = q_1 \mathbf{x}_1$ must be integral because $q_1$ is. Now assuming the induction holds for some $t \geq 1$, then

$$\mathbf{s}_{t+1} = q_{t+1} \mathbf{x}_{t+1} = q_{t+1} (\mathbf{x}_t + \eta_t (\mathbf{v}_t^+ - \mathbf{v}_t^-)) = \frac{q_{t+1}}{q_t} \mathbf{z}_t, \quad \text{where} \quad \mathbf{z}_t := \mathbf{s}_t + \mathbf{v}_t^+ - \mathbf{v}_t^-. \tag{38}$$

Consider three cases noting that both $\mathbf{v}_t^+$ and $\mathbf{v}_t^-$ are in $\{0, 1\}^n$:

- If $x_t(i) = 0$, then $v_t^-(i) = s_t(i) = 0$, and so $z_t(i) \in \{0, 1\}$.

- If $x_t(i) = 1$, then $v_t^-(i) = 1$ and $s_t(i) = q_t$. So $0 \leq z_t(i) \leq q_t + 1 - 1 = q_t$.

- If $x_t(i) \in (0, 1)$, then $s_t(i) \in [1, q_t - 1]$. So $0 \leq z_t(i) \leq q_t - 1 + 1 = q_t$.

To summarize, in all these cases, $x_{t+1}(i) = z_t(i) / q_t \in [0, 1]$, and $z_t(i)$ is obviously integral. Therefore, $\mathbf{s}_{t+1} = \frac{q_{t+1}}{q_t} \mathbf{z}_t$ is integral as $\frac{q_{t+1}}{q_t}$ is integral. $\qquad\square$

*Proof of Lemma 2.* To present a unified proof, we do not consider the phase of $t < n_0$ and $t \geq n_0$ separately. When $t < n_0$ we can equivalently set $\gamma_t = 1$ and let AFW-1 always take a FW step up to step $n_0$. We prove by induction that $\mathbf{s}_t := q_{t-1}\mathbf{x}_t$ is integral in all coordinates and $\mathbf{x}_t \in [0, 1]^n$. When $t = 1$, since $\mathbf{x}_1$ is an extreme point, it must lie in $\{0, 1\}^n$. Then $\mathbf{s}_1 = q_0\mathbf{x}_1 = \mathbf{x}_1$ must be integral because $q_0 = 1$. Now assuming the induction holds for some $t \geq 1$, then

$$\mathbf{s}_{t+1} = q_t\mathbf{x}_{t+1} = \begin{cases} q_t\left(\frac{q_{t-1}}{q_t}\mathbf{x}_t + \frac{1}{q_t}\mathbf{v}_t^+\right) = 2^s q_{t-1}\mathbf{x}_t + \mathbf{v}_t^+ = 2^s\mathbf{s}_t + \mathbf{v}_t^+, & \text{if step } t \text{ is FW} \\ q_t\left(\frac{q_t+1}{q_t}\mathbf{x}_t - \frac{1}{q_t}\mathbf{v}_t^-\right) = 2^s q_{t-1}\mathbf{x}_t - \mathbf{v}_t^- = 2^s\mathbf{s}_t - \mathbf{v}_t^-, & \text{if step } t \text{ is away} \end{cases}.$$

(39)

So in both cases, $\mathbf{s}_{t+1}$ is integral by induction. Obviously $\mathbf{x}_{t+1} \in [0, 1]^n$ if step $t$ is FW. When step $t$ is away, consider three cases noting that both $\mathbf{v}_t^+$ and $\mathbf{v}_t^-$ are in $\{0, 1\}^n$:

- If $x_t(i) = 0$, then $v_t^-(i) = 0$ and $s_t(i) = 0$. Thus $s_{t+1}(i) = 0$ and $x_{t+1}(i) = 0$.

- If $x_t(i) = 1$, then $v_t^-(i) = 1$ and $x_{t+1}(i) = 1$.

- If $x_t(i) \in (0, 1)$, then $s_t(i) \in [1, q_{t-1} - 1]$. So

$$\mathbf{x}_{t+1}(i) = \left(1 + \frac{1}{q_t}\right)\mathbf{x}_t(i) - \frac{1}{q_t}\mathbf{v}_t^-(i) = \frac{1}{q_t}\left(2^s q_{t-1}\mathbf{x}_t(i) - \mathbf{v}_t^-(i)\right)$$

$$\begin{cases} \geq \frac{1}{q_t}(2^s - 1) \geq 0 \\ \leq \frac{1}{q_t}2^s(q_{t-1} - 1) = \frac{2^s(q_{t-1}-1)}{2^s q_{t-1}-1} \leq 1 \end{cases}.$$

□

*Proof of Lemma 3.* By Eq 4 of [4], we have $h_{t+1} \leq (1 - \eta_t)h_t + \eta_t^2 M_2$. Clearly $h_1 \leq M_2$ and $h_2 \leq M_2$. Assume the result holds for some $t \in [2, n_0 - 1]$. Then by induction,

$$h_{t+1} \leq \frac{t-1}{t}h_t + \frac{1}{t^2}M_2 \leq \frac{t-1}{t}\frac{3}{t}M_2\log t + \frac{1}{t^2}M_2 \leq \frac{3}{t+1}M_2\log(t+1). \qquad \square$$

*Proof of Lemma 4.* b) Since $\gamma_{t+1}^{-1} - \gamma_t^{-1}$ increases in $t$, so

$$\gamma_{t+1}^{-1} - \gamma_t^{-1} \geq 1 \quad \Leftrightarrow \quad \gamma_{n_0+1}^{-1} - \gamma_{n_0}^{-1} \geq 1 \tag{40}$$

$$\Leftrightarrow \quad (1 - c_1)^{1-n_0} \geq \frac{M_1^2 c_0}{\theta^2 M_2^2}(1 - (1 - c_1)^{-0.5})^{-2} \approx \frac{M_1^2 c_0}{\theta^2 M_2^2}\frac{4}{c_1^2} \tag{41}$$

$$\Leftrightarrow \quad \frac{c_0 n_0}{3M_2 \log n_0} \geq \frac{M_1^2 c_0}{\theta^2 M_2^2}\frac{4}{c_1^2} \quad \Leftrightarrow \quad \frac{n_0}{\log n_0} \geq \frac{12M_1^2}{\theta^2 M_2 c_1^2}. \tag{42}$$

If we approximate $n_0/\log n_0$ by $n_0$, then using $n_0 c_1 \approx 1$ we obtain

$$c_1 \geq \frac{12M_1^2}{M_2} \quad \Leftrightarrow \quad \frac{M_1^2}{M_2}\frac{\theta-4}{4\theta^2} \geq \frac{12M_1^2}{\theta^2 M_2 c_1^2}. \tag{43}$$

This holds as equality since $\theta = 52$. If we do not ignore the log term, then note that for $n_0/\log n_0 = a$, we only need to set $n_0 = a \cdot \log a \cdot \log\log a\ldots$, until the log of the log (and so on) is less than 1. Since $\log a = \log(12M_1^2/(\theta^2 M_2 c_1^2))$ can be considered as a small *universal* constant, the subsequent proof only needs to be scaled slightly.

a) Obviously $\gamma_t$ is decreasing and hence it suffices to show $\gamma_{n_0} \leq 1$. By using (41), we get

$$\gamma_{n_0} = \frac{M_1}{\theta M_2}\sqrt{c_0}(1 - c_1)^{(n_0-1)/2} \leq \frac{M_1}{\theta M_2}\sqrt{c_0} \cdot \frac{\theta M_2 c_1}{2M_1\sqrt{c_0}} = \frac{c_1}{2} < 1. \tag{44}$$

c) By definition, $\eta_t = q_t^{-1} \leq 1/\lceil\gamma_t^{-1}\rceil \leq \gamma_t$. To show $\frac{1}{4}\gamma_t \leq \eta_t$, it suffices to show $\eta_t^{-1} \leq 2\lceil\gamma_t^{-1}\rceil$ because $\lceil\gamma_t^{-1}\rceil \leq 2\gamma_t^{-1}$ ($\gamma_t \leq 1$). To prove $\eta_t^{-1} \leq 2\lceil\gamma_t^{-1}\rceil$, we first note that it holds for $t = n_0$ because $\eta_{n_0}^{-1} = n_0 = \lceil c_1^{-1}\rceil \leq 2\lceil 2c_1^{-1}\rceil \leq 2\lceil\gamma_{n_0}^{-1}\rceil$ (the last inequality is by (44)). Assuming $q_t = \eta_t^{-1} \leq 2\lceil\gamma_t^{-1}\rceil$ holds for some $t \geq n_0$, we next perform induction on $t + 1$ by considering four cases.

- $s = 0$ and the step is FW. Note $q_{t+1} = q_t + 1 \leq 2 \lceil \gamma_t^{-1} \rceil + 1 \leq 2 \lceil \gamma_{t+1}^{-1} \rceil - 1$. The last inequality is because $\gamma_{t+1}^{-1} - \gamma_t^{-1} \geq 1$ (in b) implies $\lceil \gamma_{t+1}^{-1} \rceil - \lceil \gamma_t^{-1} \rceil \geq 1$.

- $s = 0$ and the step is away. By induction, $q_{t+1} = q_t - 1 \leq 2 \lceil \gamma_t^{-1} \rceil - 1 < 2 \lceil \gamma_{t+1}^{-1} \rceil$ because $\gamma_t$ is decreasing in $t$.

- $s \geq 1$ and the step is FW. By definition, $2^{s-1} q_t + 1 < \lceil \gamma_{t+1}^{-1} \rceil$. Thus $q_{t+1} = 2^s q_t + 1 \leq 2 \lceil \gamma_{t+1}^{-1} \rceil - 1 < 2 \lceil \gamma_{t+1}^{-1} \rceil$.

- $s \geq 1$ and the step is away. By definition, $2^{s-1} q_t - 1 < \lceil \gamma_{t+1}^{-1} \rceil$. Since both sides of the inequality are integers, this means $2^{s-1} q_t - 1 \leq \lceil \gamma_{t+1}^{-1} \rceil - 1$. Thus

$$q_{t+1} = 2^s q_t - 1 \leq 2 \lceil \gamma_{t+1}^{-1} \rceil - 1 < 2 \lceil \gamma_{t+1}^{-1} \rceil. \qquad \square$$

*Proof of Example 5.* In fact let $n = 2^q$ for some positive integer $q$, and $\mathbf{x}_1 = \epsilon \sum_{i=1}^n i \mathbf{e}_i$. Then it is easy to see that $\mathbf{x}_1 = H \cdot \frac{n\epsilon}{n-1} (2^0, 2^1, \ldots, 2^{q-1})^\top$, where $H$ is a $2^q \times q$ matrix whose rows enumerate all the binary assignments of $q$ bits. So $\mathbf{x}_1$ is the convex combination of $q + 1$ vertices ($\mathbf{0}$ included). It turns out that AFW-2 will first pick an away direction $\mathbf{1}$, then another away direction $\mathbf{1} - \mathbf{e}_1$, followed by $\mathbf{1} - \mathbf{e}_1 - \mathbf{e}_2$, etc. $\qquad \square$

# B  Details of Updates for AFW and PFW on SVM

Given the gradient $\mathbf{g}$, the FW and away directions can be computed efficiently. The FW direction needs to solve

$$\min_{\mathbf{v}} \mathbf{v}^\top \mathbf{g}, \quad s.t. \quad \mathbf{v} \in [0, 1]^n, \quad \sum_{i \in P} v_i = \sum_{j \in N} v_j, \tag{45}$$

where $P$ and $N$ are the index set of positive and negative examples respectively. To solve it, one just needs to sort $\{g_i : i \in P\}$ and $\{g_j : j \in N\}$ separately in a decreasing order, *e.g.* $g_{i_1}^+ \geq g_{i_2}^+ \geq \ldots$. Then we just need to find the smallest $k$ such that $g_{i_k}^+ + g_{j_k}^- < 0$, or $|P|$, or $|N|$, whichever is the smallest. The away direction is similar, and $\mathcal{P}(\mathbf{x}_t)$ simply forces some $v_i$ to be either 0 or 1.

The final line search can be written as $\min_{\eta \geq 0} \frac{1}{2} \eta^2 \mathbf{d}_t^\top Q \mathbf{d}_t + \eta \mathbf{x}_t^\top Q \mathbf{d}_t - \eta \frac{1}{C} \mathbf{1}^\top \mathbf{d}_t$, s.t. $\mathbf{x}_t + \eta \mathbf{d}_t \in [0, 1]^n$. We have shown above how to compute $Q \mathbf{d}_t$ efficiently. The constraint effectively restricts $\eta$ to an interval, and so the optimal $\eta$ for the quadratic objective can be found in closed form.

## B.1  Computational efficiency per iteration.

Denote $\mathbf{z} = [\mathbf{u}; \mathbf{v}]$. At each step $t$ of AFW and PFW, one needs to compute the gradient in $\mathbf{z}$, which is exactly $Q\mathbf{z}_t$. Suppose the part corresponding to $\mathbf{u}$ is $\mathbf{g}_u$. Then the FW direction needs to solve $\min_{\mathbf{u} \in \mathcal{P}_K} \mathbf{u}^\top \mathbf{g}_u$. This can be easily solved by finding the largest $K$ coordinates of $\mathbf{g}_u$. For away-step, it simply clamps some elements in $\mathbf{u}$ to 0 or 1. So $\mathbf{d}_t$ in AFW and PFW have at most $2K$ and $4K$ nonzeros respectively, and it costs $O(nK)$ time to update the gradient. The scheme is very similar to that for dual SVM.

## B.2  Translation between RC-Hull (23) and SVM Dual (20)

Theorem 4.4 of [21] showed how to convert the optimal $(\mathbf{u}, \mathbf{v})$ of RC-Hull to the optimal solution of SVM-Dual. In short, one first computes $\boldsymbol{\theta}$ of RC-Margin by $\boldsymbol{\theta} = \frac{1}{K}(A\mathbf{u} - B\mathbf{v})$. Then fixing $\boldsymbol{\theta}$, we can find the optimal $\alpha$ and $\beta$ for RC-Margin with a closed form (see Appendix B.3). Next we compute a scaling factor $\delta = \frac{2}{\alpha + \beta}$, and the $C$ can be recovered by $C = \frac{\delta}{K}$. Finally the optimal $\mathbf{x}$ of SVM-Dual is simply $(\mathbf{u}^\top, \mathbf{v}^\top)^\top$, assuming all positive examples are indexed before negative examples. As a result, the number of support vector in SVM-Dual is exactly the number of zeros in the optimal solution of RC-Hull.

## B.3  Finding $\alpha$ and $\beta$ given $\theta$ in RC-Margin

Given $w$ to find the biases $\alpha$ and $\beta$ we need to solve the following optimization problem:

$$\min_{\alpha,\beta,\xi,\eta} D(\sum_i \xi_i + \sum_i \eta_i) - \alpha + \beta$$

$$\textbf{s.t.} \quad A_i w - \alpha + \xi_i \geq 0 \quad \xi_i \geq 0$$
$$- B_i w - \beta + \eta_i \geq 0 \quad \eta_i \geq 0$$

**Solution**. Note that $\alpha$ and $\beta$ are decoupled in the above equation so we're going to solve them separately:

$$\min_{\alpha,\xi} \quad D\sum_i \xi_i - \alpha$$

$$\textbf{s.t.} \quad \xi_i \geq \alpha - a_i \quad \xi_i \geq 0,$$

where $a_i = A_i w$ are constants. WOLG, assume $a_1 \leq a_2 \leq ... \leq a_n$. Suppose $\alpha^*$ is the solution to this problem. We can easily show that $a_1 \leq \alpha^*$ [2]. Suppose k is the largest index that $a_k \leq \alpha^*$. Hence, we'll have:

$$\xi_i^* = \begin{cases} \alpha^* - a_i & \text{if } i \leq k \\ 0 & \text{if } i > k \end{cases}.$$

Thus, we have:

$$D\sum_i \xi_i^* - \alpha^* = D\sum_{i=1}^k (\alpha^* - a_i) - \alpha^* = (kD - 1)\alpha^* - D\sum_i^k a_i.$$

So $\alpha^*$ is minimizing this expression subject to $\alpha^* \geq a_k$. It is obvious in this case $\alpha^* = a_k$. Thus, we can write down:

$$D\sum_i \xi_i^* - \alpha^* = ((k-1)D - 1)a_k - D\sum_i^{k-1} a_i$$

So the problem is to find the k that minimizes $-(1 - (k-1)D)a_k - D\sum_i^{k-1} a_i$. As long as $k - 1 \leq \frac{1}{D}$ this expression is negative of a convex combination of $a_1, a_2, ..., a_k$ and since $a_i's$ are increasing in $k$, the expression is decreasing in $k$ as well until we reach a $k$ that $k - 1 > \frac{1}{D}$. After that point the expression is increasing in k since the coefficient of largest $a_i$ is positive. To see this, consider two consecutive expressions

$$((k-1)D - 1)a_k - D\sum_i^{k-1} a_i < (kD - 1)a_{k+1} - D\sum_i^k a_i$$

$$\Leftrightarrow \quad (kD - 1)a_k < (kD - 1)a_{k+1}.$$

So as long as $kD < 1$ or $k < \frac{1}{D}$, the expression is decreasing in k and when $k > \frac{1}{D}$ it is increasing so the minimum is where $k = \lceil \frac{1}{D} \rceil$. If $k = \frac{1}{D}$, the expression is the same for $k$ and $k + 1$ (In this case any $a_k \leq \alpha^* \leq a_{k+1}$ is a solution to this problem).

## Footnotes

[2]Suppose $\alpha^* < a_1$. Therefore, $\xi_i^* = 0$ for all $i$. $D\sum_i \xi_i^* - \alpha^* = -\alpha^* > -a_1$.