[Reviews · NeurIPS 2017]

Reviewer 1



The main result of the paper is an analysis of the away-step Frank-Wolfe showing that with line search, and over a polyhedral domain, the method has linear convergence. This is somewhat restrictive, but more general than several recent related works, and while they analyze some delicate fixed step-size methods, their main point seems to be that they can analyze the line search case which is important since no one uses the fixed step-size methods in practice (since they depend on some unknown parameters). Some parts of the paper are written well, and it appears the authors are highly competent. A good amount of proofs are in the main paper, and the appendix supports the paper without being overly long. However, for someone like myself who is aware of Frank-Wolfe but not familiar with the pairwise or away-step FW and their analysis, nor with the simplex-like polytopes (SLP), the paper is quite hard to follow. Section 3.1 and 3.2 discuss a metric to quantify the difficulty of working over a given polytope, and this is central to the paper and would benefit from a more relaxed description. Also, much of the paper is concerned with the SLP case so that they can compare with previous methods, and show that they are comparable, even if this case (with its special stepsize) is not of much practical interest, so this adds to the complexity. And overall, it is simply a highly technical paper, and (like many NIPS papers) feels rushed in places -- for example, equation (4) by itself is not any kind of valid definition of H_s, since we need something like defining H_s as the maximum possible number such as (4) is true for all points x_t, all optima x^*, and all possible FW and away-directions. The numerical experiments are not convincing from a practitioners point-of-view, but for accompanying a theory paper, I think they are reasonable, since they are designed to illustrate key points in the analysis (such Fig 2 showing linear convergence) rather than demonstrate this is a state-of-the-art method in practice. After reading the paper, and not being a FW expert, I come away somewhat confused about what we need to guarantee linear convergence for FW. We assume strong convexity and smoothness (Lipschitz continuity of the gradient), and a polyhedral constraint set. At this point, other papers need the optimal solution (unique, via strong convexity) to be away from the boundary or similar, but this paper doesn't seem to need that. Can you clarify this? Overall, I'm impressed, and mildly enthusiastic, but I think some of the parts I didn't understand well are my fault and some could be improved by a better-expressed paper, and overall I'm somewhat reserving judgement for now. Minor comments: - Line 26, "Besides," is not often used this way in written material. - Line 36, lots of commas and the "e.g." make this sentence hard to follow - Line 39, "empirical competency" is not a usual phrase - Line 63, "SMO" is not defined, and I had to look at the titles of refs [19,20] to figure it out - Eq (1), it's not clear if you require the polytope to be bounded (some authors use the terminology "polytope" to always mean bounded; maybe this is not so in the literature you cite, but it's unclear to a general reader). Basic FW requires a compact set, but since your objective is assumed to be strongly convex, it is also coercive, so has bounded level sets, so the FW step is still well-defined, and I know FW has been analyzed in this case. So it does not seem to be necessary that the polytope is bounded, and I was very unsure if it was or not (had I been able to follow every step of the proofs, this would probably have answered itself). - Line 80, "potytope" is a typo - Eq (4) is not a definition. This part is sloppy - Eq (5) is vague. What does "second_max" mean? The only definition that makes sense is that if we define I to be the set of all indices that achieve the max, then the second max is the max over all remaining indices, though this could be an empty set, so then the value is not defined. Another definition that doesn't seem to work in your case is that we exclude only one optimal index. It was hard to follow the sentence after (5), especially given the vagueness of (5). - Line 145 defines e_1, but this should be earlier in line 126. - Section 3.2 would benefit from more description on what these examples are trying to show. Line 148 says "n which is almost identical to n-1", which by itself is meaningless. What are you trying to express here? I think you are trying to show the lower bound is n-1 and the upper bound is n? - Line 160, is there a downside to having a slack variable? - Line 297, "not theoretical" is a typo. - Line 311, "Besides," is not used this way. == Update after reading authors' rebuttal == Thanks for the updates. I'm still overall positive about the paper.

Reviewer 2



The authors consider a frank wolfe method with away steps, and derive linear convergence bounds for arbitrary polytopes, for smooth and strongly convex loss functions. They introduce a novel notion of the geometrical complexity of the constraint set, and show that this complexity measure can be computed relatively easily (or bounded). They also derive statistical rates for kernel SVM, and show improved performance when compared to the SMO method. Detailed comments below: - In general, is there some relationship between H_s and Pw? FOr example, in Table 1 last row, it is hard to tell what is better since there is no way of comparing 1/Pw with n H_s - line 122: what is ext(P) ? - Section 3.2 essentially implies that the bounds are tight up to constant factors. Please mention this clearly. - Theorem 1 and Lemma 2: I don't get the "must be feasible" part. Don't Frank Wolfe methods always ensure that the iterates are feasible (via the convex combination of the new atom and old iterate)? - I think the paper will benefit from providing wall clock time for the experiments as well. Ideally for large scale applications, the time the methods take to reach an optimum is more important. - On the same note as above, could you also compare to other methods that solve the SVM dual (libSVM, divide and conquer methods)? Again, these schemes have shown to be very scalable. Or you need to make clear why SMO is a valid choice to compare against.

Reviewer 3



This paper presents extensions of existing variants of Frank-Wolfe or conditional gradient methods; in particular, it extends the performance (convergence rate) of the Pairwise-FW variant of Garber and Meshi 2016 to handle more general polytope constraints, by using an away-step update. For the analysis of this algorithm variant, a new condition number of the feasible set is introduced that better captures the sparsity of the solution. Introducing a different notion of condition number Hs is described as one of the contributions. Can the authors present some more intuition and geometric interpretations, and more examples (other than simplex and its convex hull) where Hs is easy to bound nontrivially and reveals important conditioning information? Other comments: 1. Mention in the introduction what AFW stands for 2. Can you give intuition for constants in eps (10) and (11)? 3. It’ll be helpful to have a list of parameters that appear throughout (e.g., \alpha, \beta, D, H_s, …) mentioning if they are problem parameters or algorithm parameters, and what they depend on. It is helpful to clarify the significance of the results in this paper to decide whether they meet the high bar at NIPS. As is, I would place the paper as a (weak) accept.